# Contribution of Reliable Chromatographic Data in QSAR for Modelling Bisphenol Transport across the Human Placenta Barrier

**DOI:** 10.3390/molecules28020500

**Published:** 2023-01-04

**Authors:** Clémence A. Gély, Nicole Picard-Hagen, Malika Chassan, Jean-Christophe Garrigues, Véronique Gayrard, Marlène Z. Lacroix

**Affiliations:** 1ToxAlim (Research Centre in Food Toxicology), National Research Institute for Agriculture, Food and Environment (INRAE), National Veterinay School of Toulouse (ENVT), University of Toulouse, 31076 Toulouse, France; 2Therapeutic Innovations and Resistances (INTHERES), National Research Institute for Agriculture, Food and Environment (INRAE), National Veterinay School of Toulouse (ENVT), University of Toulouse, 31076 Toulouse, France; 3Molecular Interactions and Chemical and Photochemical Reactivity Laboratory (IMRCP), University of Toulouse, 31062 Toulouse, France

**Keywords:** bisphenols, QSAR, human placental transfer, endocrine disruptor, chromatographic descriptors

## Abstract

Regulatory measures and public concerns regarding bisphenol A (BPA) have led to its replacement by structural analogues, such as BPAF, BPAP, BPB, BPF, BPP, BPS, and BPZ. However, these alternatives are under surveillance for potential endocrine disruption, particularly during the critical period of fetal development. Despite their structural analogies, these BPs differ greatly in their placental transport efficiency. For predicting the fetal exposure of this important class of emerging contaminants, quantitative structure-activity relationship (QSAR) studies were developed to model and predict the placental clearance indices (CI). The most usual input parameters were molecular descriptors obtained by modelling, but for bisphenols (BPs) with structural similarities or heteroatoms such as sulfur, these descriptors do not contrast greatly. This study evaluated and compared the capacity of QSAR models based either on molecular or chromatographic descriptors or a combination of both to predict the placental passage of BPs. These chromatographic descriptors include both the retention mechanism and the peak shape on columns that reflect specific molecular interactions between solute and stationary and mobile phases and are characteristic of the molecular structure of BPs. The chromatographic peak shape such as the asymmetry and tailing factors had more influence on predicting the placental passage than the usual retention parameters. Furthermore, the QSAR model, having the best prediction capacity, was obtained with the chromatographic descriptors alone and met the criteria of internal and cross validation. These QSAR models are crucial for predicting the fetal exposure of this important class of emerging contaminants.

## 1. Introduction

Bisphenol A (BPA), which was widely used in everyday products, has been prohibited in food packaging in several countries because of its endocrine disrupting properties. As a result, structurally similar phenolic compounds such as BPAF, BPAP, BPB, BPF, BPP, BPS, and BPZ (Figure 1) have gradually replaced BPA in the polymer industry. Because of the wide use of these analogues in food packaging and personal care products [1,2,3], human exposure to these compounds is ubiquitous, as demonstrated by their detection in human urine [1,4,5]. These analogues may display endocrine disrupting properties similar to those of BPA [6,7,8,9,10] particularly during the critical period of pregnancy [11,12,13]. In vivo studies performed in pregnant sheep showed that, similar to BPA, BPS and BPF can also cross the placenta [14,15,16]. Recently, ex vivo human placental perfusion carried out to assess the placental transfer of 15 bisphenols (BPs) has shown that despite their structural similarities, the efficiency of placental transport differs greatly among BPs, as reflected in the large range of their clearance indices, from 0.065 to 0.842 [17]. Indeed, while BP4-4, BPAP, BPE, BPF, 3-3BPA, BPB, and BPA cross the placenta by passive diffusion [14], the materno-fetal placental transfer of BPS and BPFL is very limited and may involve membrane efflux transporters [17,18].

The physicochemical properties of the molecules, such as molecular weight, degree of ionization (pKa), and lipophilicity (LogP), are considered possible factors that determine the extent of placental passage and the mechanism of transfer [19]. In this context, several studies have predicted the maternal-fetal transfer rates of many drugs or organic compounds through quantitative-structure activity relationship (QSAR) modelling [20,21,22,23]. These models are mostly based on molecular descriptors in order to establish a relationship between the biological activities or the toxicological responses of molecules and their molecular, physicochemical, and structural properties [24]. However, these numerical descriptors are calculated from the two- and/or three-dimensional structures of molecules and do not take into account the dynamic interactions with their biological targets [25]. For BP analogues, the physicochemical parameters generally involved in placental transfer (pKa within 8.5–9, LogP within 2–7, and molecular weight (MW) below 600 Da) would imply that their potential passive diffusion across the placenta would be high [17,20,21,22]. It is, therefore, anticipated that the placental transfer efficiency of these structurally related BPs will be hardly predicted by descriptors based on molecular structure alone. In many cases, experimental descriptors such as chromatographic retention have been added to molecular ones to improve the QSAR models [26,27]. Indeed, the chromatographic retention depends on the interaction of the compound of interest, i.e., its structure, with the stationary phase and the mobile phase. Thus, chromatographic retention of a molecule depends strongly on both its physicochemical properties and its environment. To the best of our knowledge, few studies have investigated the use of chromatographic descriptors alone in QSAR modelling, and all of them have only considered retention parameters [28,29,30].

The aim of the present study was to evaluate the reliability of the QSAR model based on chromatographic descriptors compared to classical molecular descriptors to predict the placental clearance of BPs, using the placental clearance indices determined in a previous study [17] as the biological output. To that end, the set of 15 selected BPs was eluted on thirteen chromatographic columns using two elution solvents. We then built a QSAR model based either on classical molecular descriptors of the fifteen bisphenols or on chromatographic descriptors, such as retention and peak shape parameters, or on both molecular and chromatographic descriptors. Their performances for predicting the BP placental transport efficiency were verified by cross validation procedures. This QSAR approach, by allowing the prediction of a key parameter, i.e., placental clearance, should provide new insight into screening of this important class of emerging BPs and assessing the hazards of BPA substitution in a vulnerable developing fetus.

## 2. Results

### 2.1. Data Collection

#### 2.1.1. Clearance Indices

Clearance indices (CI) of the 15 BPs perfused in cocktail were determined on five placentae using the ex vivo human perfused placenta model described in our previous study [17], as shown in Table 1. CI data were used as outcomes due to their ability to eliminate inter-placental variation by standardizing BP clearance to the reference compound antipyrine. These BPs were classified into two groups according to their mechanism of placental passage, by passive diffusion for seven BPs (similar to antipyrine) or the limited transport for eight BPs [17]. This data set obtained in the same experimental conditions was relatively distributed in a wide range of placental CI and is, thus far, suited for the QSAR modelling of the ability of these newly-introduced BPs to cross the placenta [22]. Moreover, the use of individual data of CI instead of the mean CI of both experimental and biological variabilities needs to be taken into account and, thus, improves the predictive capacity of the model [31].

#### 2.1.2. Molecular Descriptors

In total, 50 molecular descriptors were calculated for each BP with different algorithms available in the ChemDraw Pro 17.1 software. Among these 50 descriptors, 16 concern physicochemical properties (e.g., pKa, LogP), 14 are topological descriptors (e.g., Balaban index, polar surface), and 20 are involved with thermodynamic and electronic properties.

#### 2.1.3. Chromatographic Descriptors

Figure 2 shows that the retention and the elution order of BPAF, BPAP, and 3-3BPA on the C18 column change according to the elution solvent, whether AcN or MeOH, (Figure 2) and the BPS peak shape is larger with MeOH than with AcN. Peak asymmetry, tailing factor, peak width (5%), and k′ vs. BPA were determined with the two organic solvents and for the 13 columns, resulting in a total of 104 descriptors per BP (Appendix A).

### 2.2. QSAR Modelling

First, the ability of the models with molecular descriptors obtained in silico to predict the BP CI was investigated. Secondly, the chromatographic descriptors were added to the molecular descriptors to assess whether this combination could improve the prediction ability of the model. Finally, a model using only chromatographic descriptors was developed and compared to those using molecular descriptors.

#### 2.2.1. Molecular Descriptors in the QSAR Model for Predicting Placental Passage

Among the 50 normalized molecular descriptors (Appendix A), eleven descriptors gave a weight greater than twice the median of the weights using artificial neural network (ANN) learning and were selected as significant descriptors in the prediction of CI. Three of them were physicochemical descriptors (partition coefficients: LP2 and CD2, and a melting point CPP10), two were steric descriptors (Connolly Molecular Area CPS2, Connolly Solvent Excluded Volume CPS3), and two were topologic descriptors (Cluster count MT2 and Polar Surface Area MT5). The other four were involved in electrostatic and thermodynamic interactions (heat of formation DE1, total energy DE2, cosmo area DE5, and ionization potential DE7). Then, using these eleven descriptors, twenty-eight combinations were selected to build the model, considering low pair correlations among the variables (determination coefficient < 0.5). Five of these combinations had a regression R² > 0.7 and significant parameters in the multilinear regression (MLR). The final model, with significant and not correlated descriptors, was obtained with the following four molecular descriptors: LogP (LP2), Polar Surface Area (MT5), the heat of formation (DE1), and the total energy (DE2), and by using Equation (1) with a weighting by 1/Y:CI = 0.46 + 0.23 × LP2 − 0.08 × MT5 − 0.30 × DE1 + 0.41 × DE2(1)

Regression-adjusted CI and leverage are reported in Appendix A. The criterion of the regression and of the cross validation are acceptable with a mean coefficient of regression determination (R^2^) of 0.77 and mean cross-validated correlation coefficient (Q^2^) higher than 0.70, respectively. However, the mean correlation coefficient of leave-many-out (LMO) validation (QLMO²) and mean correlation coefficient of concordance (CCC) for the three data sets were 0.57 and 0.64, respectively, which did not fulfill the recommended criteria for validating the predictive ability of this model (i.e., QLMO² > 0.6 and CCC > 0.85, Table 2).

Moreover, the slope of the LMO validation curve [Ypred = f(mean(Yobs))] is significantly different than 1 (Figure 3A), and for most of the BPs of the test sets, the predicted CI are underestimated by the model except for BPZ (Figure 4A), for which its predicted CI were twice as high as its observed CI (Figure 4A). The results suggest that these in-silico calculated descriptors alone are not robust enough for predicting the clearance index of a new BP.

#### 2.2.2. Combining Chromatographic Descriptors and Molecular Descriptors in the QSAR Model for Predicting Placental Passage

The 104 chromatographic descriptors were combined with the 50 molecular descriptors. Thirty-eight descriptors from these 154 descriptors were selected by ANN (Table 3), leading to 341 combinations of descriptors without correlation. Interestingly, only five molecular descriptors included in the previous molecular QSAR modelling (MT2, MT5, DE1, DE2, and DE5) were also significant in this present model, whereas the molecular descriptors related to LogP were not selected by ANN using molecular and chromatographic descriptors. The retention factors related to BPA were first selected by ANN (k′ vs. BPA, #5), but were progressively removed from the multilinear regression models because of their strong correlations with the molecular and peak shape descriptors. After the MLR assay, 50 models were chosen for their good fitting and the significance of their parameters. However, only a few of them met the LMO validation criteria.

Furthermore, the influence of BPs (leverage) on the regression differed according to the type of descriptors used. Indeed, BPAF, which had a strong influence using molecular descriptors, was no longer leveraged but was replaced by BP44, resulting in a change in the arbitrary choice of BPs in the test set 1 for LMO validation [32]. The final model included the following parameters: Connolly Solvent Excluded Volume (CPS3), cluster count (MT2), and LUMO energy (DE9) as molecular descriptors, and shape peak parameters (width #2, asymmetry #7, and tailing factor #8) on the biphenyl and BEH C8 column with AcN elution (RCBA8 and C8A2) and on the PFP column with MeOH as the organic modifier (PFPM2 and PFPM7). The equation of the multilinear regression, assigning a weight 1/Y, was:CI = 0.48 − 0.03 × CPS3 − 0.23 × MT2 + 0.33 × DE9 − 0.25 × C8A2 + 0.37 × RCBA8 − 0.27 × PFPM2 + 0.23 × PFPM7(2)

Regression-adjusted CI and leverage are reported in Appendix A. The R² of the regression was 0.85, the Q² of the cross validation was 0.82, and the Bayesian information criterion (BIC) was lower than that of the model with molecular descriptors. However, the QLMO² of the LMO validation was lower than 0.5 (Table 2).

As shown in Figure 3B, the correlation curve (straight line) was systematically below the perfect curve (dot line) and the predicted CI for the BPs having low CI were misestimated (Figure 4B). The combination of chromatographic and molecular descriptors was not able to improve the prediction capacity of the QSAR model based solely on molecular descriptors.

#### 2.2.3. Chromatographic Descriptors in the QSAR Model for Predicting Placental Passage

Among the 104 descriptors experimentally obtained by liquid chromatography, the ANN selected 17 chromatographic descriptors with weights higher than twice the median of all descriptors’ weight (Table 4). Eight chromatographic descriptors were also significant in the QSAR model based on both molecular and chromatographic descriptors (Table 3). Twenty-four models based on uncorrelated descriptors were identified, resulting in the selection of eight models with significant descriptors for predicting CI.

The final model was obtained using six chromatographic descriptors with a weighting by 1/Y according to the following Equation (3):CI = 0.48 − 0.66 × T3A2 − 0.05 × PFPA7 + 0.07 × CC18A7 − 0.50 × C8M7 − 0.14 × PFPM7 + 0.38 × CNM8(3)

Regression-adjusted CI and leverage of Equation (3) are reported in Appendix A. All of these descriptors referred to peak shape parameters, and half were eluted with AcN and half with MeOH. Three of them were eluted on a hydrophobic phase (HSS T3, C18, and C8), two on a fluorophenyl phase (PFP), and one on a phase containing a polar group (CN). The asymmetry on the PFP column with MeOH elution (PFPM7) was the only descriptor in common with the multilinear regression based on these molecular and chromatographic descriptors (Equation (2)). This model has the lowest BIC value and the R² of the regression and the Q² of the cross-validation were 0.84 and 0.81, respectively. Moreover, the LMO validation values (QLMO² = 0.73 and CCC = 0.81) almost met the recommended criteria (Table 2), [33]. The Spearman’s correlation coefficient (R² spearman) was higher than 0.8 (Table 2), attesting to the ability of the model to classify CI according to the BPs. In addition, the mean square error of the calibration (RMSEC), cross-validation (RMSECV), and prediction (RMSEP) were low, between 0.11 and 0.13 (Table 2). The correlation curve (straight line, Ypred = f(mean(Yobs))) was close to the perfect line (dotted line, Figure 3C) with an intercept not significantly different from 0 and a slope close to 1. The predicted CI of BPs were grossly well estimated, with predicted CI included in standard deviation intervals of experimented values, except for BPE, BPF, and BPAF, which were slightly overestimated, and BPP, which was slightly underestimated (Figure 4C).

## 3. Discussion

This work aimed at developing a predictive model of placental transfer for newly emerging BPs. This approach was based on QSAR modeling of the CI of 15 BPs previously evaluated using ex vivo human placental perfusion [17]. BPs were selected because of their occurrence in foodstuffs [34,35], their production volume [36], and the diversity of their structures. In the present study, we have shown that a QSAR model based on chromatographic parameters provides more reliable predictions of BP placental transfer than QSAR models based on molecular descriptors alone or in combination with chromatographic descriptors.

The classical QSAR model based on molecular descriptors (Equation (1)) showed that physicochemical, topological, thermodynamic, and electronic parameters (LogP, total energy, polar surface area, and heat of formation) were able to influence the placental passage of this family of emerging BPs. Among physicochemical properties, the molecular weight and the lipophilicity are known to be key factors determining the passage across placenta [37,38,39]. The molecular weights of all BPs were below 600 Da and would not impact their passage across the placenta. Moreover, in a QSAR methodology using multivariate data analysis to model the placental passage of structurally diverse drugs and chemicals, the polar surface area and lipophilicity proved to be the most important factors in the model [21]. A QSAR model using multilinear analysis identified molecular weight, polarity, and also the heat of formation as important parameters to predict the materno-fetal transport of organohalogen compounds [20]. Whatever the importance of these factors, our results showed that a QSAR model based on molecular descriptors alone cannot provide a robust prediction of the placental transport for this class of emerging BPs, even if the ranking of their CI can be determined with a relatively good confidence level. It means that the placental transfer efficiency of these structurally related BPs cannot be predicted solely from their physicochemical properties determined in silico.

Chromatographic techniques offer the possibility of taking the interactions of the molecules with different columns in several conditions (pH, solvent) into account, and then better characterizing the molecular interactions with the biological system. That is the reason why chromatographic descriptors were combined with molecular ones in the QSAR models [26,27,28,29,30,40,41,42]. The most commonly used chromatographic descriptors are the retention factors that can be determined on several orthogonal stationary phases with different pH and solvents [26,28,30]. Indeed, many new reverse stationary phases were developed to extend the selectivity of chromatographic retention, involving many interactions with the column during elution [43]. The use of these experimental tools might be of great interest for QSAR modelling because these chromatographic parameters integrate several physicochemical properties of the compound. In this study, we used 13 chromatographic columns containing different stationary phases and two elution solvents, MeOH and AcN, involving different interactions with the 3D structure of the chemical.

Indeed, the organic modifier used in the gradient mode influences the fluidity of the alkyl chains of the stationary phases and then the mechanisms of molecular interactions between the analytes and the stationary phases [44], leading to modification of the chromatographic parameters. In our experiment, the use of MeOH as an organic modifier in the gradient mode favors the H-bonding and π-π interactions, while ion-dipole interactions are masked. Conversely, the use of AcN, due to the interactions of its cyano function with the stationary phase, limits the π-π interactions [45] and promotes dipole-dipole interactions [46]. The chromatographic behavior of the chemical is described not only by the retention factors but also by the peak shape parameters, such as asymmetry, tailing factors, or width at 5% [47,48], and can be linked to specific molecular descriptors [44].

Our modelling approach (Equation (2)) showed that the Connolly Solvent Excluded Volume, which is the volume enclosed in the molecule, had a weak influence on the placental passage prediction. The topological descriptor, cluster count, which reflects the complexity of the molecule [49], had a negative contribution to the CI of bisphenols. Inversely, the LUMO energy had a positive contribution. Our results are consistent with a previous QSAR model [20] showing that electronic descriptors are important parameters that may predict the ability of the molecule to cross the placenta. Surprisingly, although the goodness of fit is better, the contribution of the chromatographic descriptors does not improve the global predictive ability of our model based on the molecular descriptors. These results are not consistent with a previous study showing that the nonlinear QSAR model of prediction of gastrointestinal absorption was improved by adding the measured chromatographic descriptors to the molecular ones [27]. This discrepancy could be attributed to the QSAR model used, which was linear in our study and nonlinear in the Deconinck study, and to differences in the chromatographic parameters used in our QSAR model, which not only include the retention factors but also the peak shape parameters such as asymmetry, tailing factor, and peak width. These chromatographic parameters evaluated in the same experimental conditions reflect, in part, the mass transfer kinetics and thermodynamics of the analytes between the mobile and the stationary phase [50]. The chromatographic stationary phase selected in our model mainly involved polarizability (RCB) and π-π interactions (PFP) of the analytes. Thus, the implementation of these chromatographic descriptors led to a combination of uncorrelated descriptors having a great influence on the prediction of placental passage, which was quite different from those selected in the two QSAR model based either on molecular or chromatographic parameters. A significant influence of the thermodynamic properties of the BPs at the expense of the lipophilicity parameters is observed. In fact, the evaluation of the relationship between the molecular and the chromatographic descriptors selected in the two chromatography-based QSAR models through the ANN learning showed that the selected chromatographic descriptors are a combination of molecular descriptors selected to describe CI. Nevertheless, the four chromatographic descriptors presented in the model based on molecular and chromatographic descriptors are not directly related to LogP, unlike the chromatographic descriptors selected by the chromatography-based QSAR (Appendix A).

When we developed a QSAR model based on chromatographic descriptors alone to predict the passage of BPs across the placenta, the main chromatographic descriptors selected by the ANNs involved the peak shape, such as asymmetry, width, and tailing factor, as in the previous model. More specifically, the most important descriptors in the validated model (Equation (3)) are the width peak on the HSS T3 column with AcN as the eluent, followed by the asymmetry peak on the BEH C8 column with MeOH elution and the tailing factor on the HSS CN column with MeOH elution. Several studies have shown that residual silanols can increase the tailing factor of basic or polar compounds due to hydrogen bonding between oxygen and hydroxyl groups or ion exchange [51]. The HSS CN stationary phase is not end capped; thus, the silanols are accessible to the eluted compounds and the CN group favors the retention of polar compounds due to dipole interactions. The importance of this descriptor could reflect the involvement of H bond interactions in the transplacental passage, which was highlighted in several QSAR studies [21,22,23]. Moreover, the predictive capacity of the model obtained with the multilinear regression and based on these chromatographic descriptors alone has been verified by an extensive internal validation procedure. Therefore, chromatographic descriptors appear to be more relevant than molecular descriptors alone or in combination with chromatographic descriptors for predicting the passage of BPs across the placental barrier [33,52,53].

The main limitation of this study is the restricted number of BPs included in the models, which did not allow for an external validation, i.e., predicting the CI of BPs not included in the QSAR development process. Consequently, leave-many-out validation was performed with eleven BPs in the training set and four in the test set, and this validation was repeated three times with three training and test sets. Moreover, unlike what is usually practiced in QSAR modelling, our training sets contained five repeatedly measured CI for each bisphenol, and it was previously shown that the incorporation of experimental errors improves the predictive ability of the QSAR model [31]. Moreover, it is important to consider that BPs with a high leverage value (as BPS and BPFL) must be included in the training set to reinforce the model, and their prediction should be considered unreliable [32,54]. Indeed, whatever the QSAR model may be, BPS and BPFL have a strong influence on the regression, which explains the poor prediction of their CI when one of them was not included in the training dataset. Interestingly, the placental CI of these two BPs were significantly lower than those of the other BPs [17], and they are structurally different in terms of lipophilicity for BPS and thermodynamic for BPFL. Regarding their chromatographic behavior, BPS and BPFL showed inverse asymmetric parameters, especially on the HSS PFP column with MeOH elution, reflecting the involvement of both H-bonds and π-π interactions. In the same way, BP44, which has a significant influence on the prediction of placental passage (high leverage) in the QSAR model based on chromatographic descriptors, was excluded for prediction even if its CI could be predicted with good accuracy using QSAR based only on molecular descriptors. Conversely, BPAF could not be included in the test set in the QSAR model based only on molecular descriptors because of the great difference in its thermodynamic properties, which is related to its fluorine atoms, whereas its CI could be predicted with the QSAR model based on chromatographic descriptors. Thus, the inclusion of new compounds containing more heteroatoms in the model would extend the range of values of molecular and chromatographic descriptors, and thus improve the ability of the QSAR model to predict the CL of BPs [55].

Further investigations including the quantitative structure-retention relationship studies (QSRR) are required to link the molecular properties and the chromatographic behavior of these BPs [44,56] and, thus, to better understand the interactions involved in the placental passage of BPs.

## 4. Materials and Methods

### 4.1. Data Set

#### 4.1.1. Compounds

Bisphenol S (BPS) (purity ≥ 98%), Bisphenol A (BPA) (purity ≥ 99%), Bisphenol E (BPE) (purity ≥ 98%), 2,2-Bis(4-hydroxy-3-methylphenyl)propane (3-3BPA) (purity ≥ 97%), Bisphenol B (BPB) (purity ≥ 98%), Bis(4-hydroxyphenyl)-2,2-dichloroethylene (BPC) (purity ≥ 98%), Bisphenol BP (BPBP)(purity ≥ 98%), Bisphenol F (BPF) (purity ≥ 98%), Bisphenol FL (BPFL) (purity ≥ 97%), Bisphenol Z (BPZ) (purity ≥ 98%), 4,4′-Dihydroxybiphenyl (BP4-4) (purity ≥ 97%), Bisphenol AP (BPAP) (purity ≥ 99%), Bisphenol AF (BPAF) (purity ≥ 97%), Bisphenol P (BPP) (purity ≥ 98%), Bisphenol M (BPM) (purity ≥ 99%) were purchased from Sigma-Aldrich (Saint Louis, MO, USA).

#### 4.1.2. Molecular Descriptors

The 3D structures of these bisphenols were minimized by MMFF94 (Merck Molecular Force Field) using a semi-empirical molecular orbital package, MOPAC application, in Chem 3D Ultra software (V17.1, Perkin Elmer informatics). An RMS gradient of 0.10 was used to minimize energy with a maximum of 1000 iterations. Electrostatic (such as dipole moment, HOMO and LUMO energy, etc.), thermodynamic (such as dipole heat of formation, Gibbs free energy, etc.), steric (such as ovality, Connolly accessible area, etc.), topologic (such as polar surface area, shape attribute, etc.), and physicochemical (such as molecular weight, pKa, LogP, etc.) descriptors were calculated. The set of molecular descriptors calculated for each studied BP is detailed in the Appendix A.

#### 4.1.3. Chromatographic Descriptors

In most of the BP chromatographic studies, the BPs are eluted on a C18 column with an H2O/AcN (Acetonitrile) gradient [57,58]. Therefore, the separation of the 15 BPs was optimized using the conventional U-HPLC BEH C18 stationary phase with acetonitrile as the organic modifier (Figure 2A). Next, the same solvent percentage, flow rate, and time were applied to thirteen chromatographic columns (Table 5), either with AcN or with MeOH (methanol), in order to compare the retention and peak shape of 15 BPs in all of these chromatographic conditions. The chromatographic descriptors were calculated from the chromatographic parameters determined by the analysis of the chromatogram of each BP and injected alone in the aqueous gradient mode on 13 analytical stationary phases listed in Table 5, using two solvents as the organic modifier: methanol and acetonitrile.

Each BP solution was prepared in (H_2_O/MeOH: 50/50) at a concentration of 0.01 mg/mL. Ten microliters of each solution were injected on an Acquity UPLC system with UV detection set at 210 nm (Waters, Milford, MA, USA). The column temperature was set at 40 °C, and the BPs were eluted at 0.3 mL/min using the linear gradient mode with two organic solvents (MeOH or AcN) and water: pump A (H_2_O)/pump B (MeOH or AcN): t(0→10 min) 84% A→5% A; t(10→15 min) 5% A. The retention time, peak asymmetry (4.4%), tailing factor, and peak width (5%) were determined with Empower Suitability System software (Waters^®^). A specific retention factor k′ vs. BPA (Equation (4)) relative to BPA was calculated by Equation (4) to normalize all retentions:k′ vs. BPA = (tr_BP_ − tr_BPA_)/tr_BPA_(4)

Descriptors corresponding to peak asymmetry, tailing factor, peak width, and retention factor were identified as #7, #8, #2, and #5, respectively (Appendix A).

### 4.2. QSAR Modelling

#### 4.2.1. Variable Selection and Multi-Linear Regression

QSAR models were performed on RStudio software (Version 1.2.5001). The descriptor values were centered and scaled to the unit standard deviation of each descriptor. An artificial neural network (ANN) was created using the R package neural net (version: 1.44.2) with a sigmoid activation function and a backpropagation learning algorithm (code in Appendix A). The ANNs were adjusted by assigning a weight to each descriptor to minimize error between predicted and experimental clearance indices using a sigmoid activation function and backpropagation learning algorithm [60]. The ANN was structured with an input layer connected to each descriptor and an output layer linked to the clearance indices (CI) for each BP. The stepmax and threshold were set at 100,000 and 0.005, respectively. After the learning cycles, descriptors with a weight (absolute value) greater than twice the median of all descriptor weights were considered significant. To avoid collinearities between descriptors, a correlation matrix was calculated with the significant descriptors. When a correlation (R^2^ greater than 0.5) between two descriptors was observed, two new matrices were created, one containing the first descriptor and the other containing the other descriptor. This operation was carried out until a matrix containing uncorrelated descriptors was obtained; these descriptors were then used to build the final QSAR model. A multilinear regression (MLR) was then performed between the selected descriptors and CI for each QSAR model using Rstudio software. The regression was tested with three weightings: 1, 1/Y, and 1/Y² with Y = observed clearance index, and residuals and leverages were plotted to evaluate, by visual inspection, the quality of the regression and to choose the weighting. The models with R-squared greater than 0.7 were selected for further development. For these models, if the descriptors were not significant (*p* < 0.05), sub-models were built by subtracting the non-significant descriptors one by one, and multilinear regressions were performed until all descriptors were significant. When too many descriptors were not significant to build the sub-models manually, the stepwise method was used to obtain the best model simplification by Akaike information criterion (AIC) [61]. The goodness of fit of each model was compared using the Bayesan information criterion calculated from each of the multilinear regressions with the RStudio software.

#### 4.2.2. Data Splitting

Clearance indices for fifteen BPs were determined on five placentae, corresponding to a total of 75 values. This data set was divided into a training set (55 values, 11 BPs used for model development) and a test set of 20 values corresponding to 4 BPs randomly selected for model predictive assessment. Three prediction data sets were constructed to allow a balanced distribution of the different structures of the BPs. Each BP was allocated to one test set except BPs with high leverage in the multilinear regression model [54,62], i.e., BPS and BPFL for the model using molecular and chromatographic descriptors, BPAF for the model using only molecular descriptors, and BP44 for the model using only chromatographic descriptors. Consequently, data set 1 included BP44, BPB, 3-3BPA, and BPZ in the test set for the validation of the model based on molecular descriptors, and data set 1 included BPAF, BPB, 3-3BPA, and BPZ for the models based on chromatographic descriptors or on both types of descriptors. Data set 2 included BPP, BPM, BPAP, and BPBP in the test set, and data set 3 included BPE, BPA, BPF, and BPC in the test set. The training set was used for multilinear regression, and the corresponding test set was used for the predictive ability validation for the three data sets separately.

#### 4.2.3. Validation

Multilinear regression models were validated with the three data sets. For each data set, the training set was used to configure regression and for cross-validation. The CI of the four BPs included in the test set were predicted by data set regression. Test set predictions were used for leave-many-out validation (LMO) according to the criteria described for external validation [33,53,55,63,64]. Multilinear regression was estimated by R^2^ (coefficient of regression determination, Equation (5)) and the root mean square error of calibration (RMSEC, Equation (6)). The robustness of the QSAR model was validated by the determination of Q^2^ (cross-validated correlation coefficient, Equation (7)) and the root mean square error of cross validation (RMSECV, Equation (8)). The predictive quality of the model was estimated with Q^2^_LMO_ (Equation (9)): the correlation coefficient of LMO validation, the root mean square error of prediction (RMSEP, Equation (10)), the correlation coefficient of concordance (CCC, Equation (11)), and the Pearson squared correlation coefficient between predicted and experimental values. Finally, the Spearman’s correlation squared coefficient was determined to assess the model’s ranking capability [63]. The accepted criteria are R² > 0.65, Q2 > 0.5, QLMO² > 0.65 [52,53], and CCC > 0.85 [33]. Additionally, RMSEC, RMSECV, and RMSEP should be close and as low as possible [53].
(5)$$R^{2}=1-[\sum_{i=1}^{I}{\left(y\left(i\right)-{\hat{y}}_{c}\left(i\right)\right)}^{2}/\sum_{i=1}^{I}{\left(y\left(i\right)-\bar{y}\right)}^{2}]$$
(6)$$\mathrm{RMSEC}=\sqrt{\sum_{i=1}^{I}{\left(y\left(i\right)-{\hat{y}}_{c}\left(i\right)\right)}^{2}/\left(I-A-1\right)}$$
(7)$$Q^{2}=1-[\sum_{i=1}^{I}{\left(y\left(i\right)-{\hat{y}}_{v}\left(i\right)\right)}^{2}/\sum_{i=1}^{I}{\left(y\left(i\right)-\bar{y}\right)}^{2}]$$
(8)$$\mathrm{RMSECV}=\sqrt{\sum_{i=1}^{I}{\left(y\left(i\right)-{\hat{y}}_{v}\left(i\right)\right)}^{2}/I}$$
(9)$${\mathrm{QLMO}}^{2}=1-(([\sum_{i=1}^{n_{EXT}}{\left(y_{e}\left(i\right)-{\hat{y}}_{e}(i)\right)}^{2}]/n_{EXT})/([\sum_{i=1}^{n_{TR}}{\left(y\left(i\right)-\bar{y}\right)}^{2}]/n_{TR}))$$
(10)$$\mathrm{RMSEP}=\sqrt{\sum_{i=1}^{nEXT}{\left(\left(y_{e}\left(i\right)-{\hat{y}}_{e}(i\right)\right)}^{2}/nEXT}$$
(11)$$\mathrm{CCC}=\frac{2\times \sum_{i=1}^{n_{EXT}}\left(y_{e}\left(i\right)-{\bar{y}}_{e}\right)\left({\hat{y}}_{e}\left(i\right)-{\bar{\hat{y}}}_{e}\right)}{\left(\sum_{i=1}^{n_{EXT}}{\left(y_{e}\left(i\right)-{\bar{y}}_{e}\right)}^{2}+\sum_{i=1}^{n_{EXT}}{\left({\hat{y}}_{e}\left(i\right)-{\bar{\hat{y}}}_{e}\right)}^{2}+n_{EXT}{\left({\bar{y}}_{e}-{\bar{\hat{y}}}_{e}\right)}^{2}\right)}$$
where *y*(*i*): experimental data values of the training set, *ŷ_c_*(*i*): multilinear-regression predicted values, *ӯ*: average of the experimental values of the training set, *ŷ_v_*(*i*): cross-validation predicted values, *y_e_*(*i*): experimental data values of the test set, *ŷ_e_*(i): test predicted values, *ӯ_e_*: average of the experimental values of the test set, ${\bar{\hat{y}}}_{e}$: average of the predicted values of the LMO validation, *I*: number of samples in the training set, *A*: number of descriptors, *nEXT*: number of descriptors in the test set, *n_TR_*: number of descriptors in the training set.

## 5. Conclusions

In this study, three QSAR models based either on molecular or chromatographic descriptors alone or a combination of molecular and chromatographic descriptors were developed, and their ability to predict the placental transfer of BPs was compared. The model based on chromatographic descriptors alone showed a better predictive ability. To the best of our knowledge, this study is the first to implement new sets of chromatographic peak shape parameters in addition to the classical retention factors in QSAR modelling. These parameters allow a better prediction of placental transfer of BPs than molecular descriptors. This model may be a useful tool for a rapid screening and subsequent removal of endocrine-active emerging BPs with a high fetal exposure potential. Further investigation by means of QSRR is needed to establish a relationship between the chromatographic behaviors of BPs in the diverse separation systems and the chemical structure to better understand the interactions involved in the placental passage of BPs. Moreover, this new concept deserves to be applied to other larger molecular families as well as to other physiological processes, such as absorption or metabolism.

## Figures and Tables

**Figure 1 molecules-28-00500-f001:**
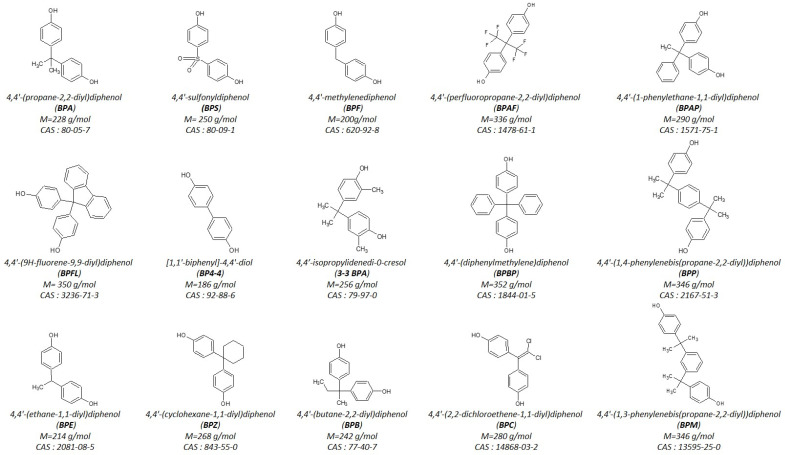
Molecular structure of the fifteen bisphenols with their respective name, abbreviation, molecular weight, and CAS number.

**Figure 2 molecules-28-00500-f002:**
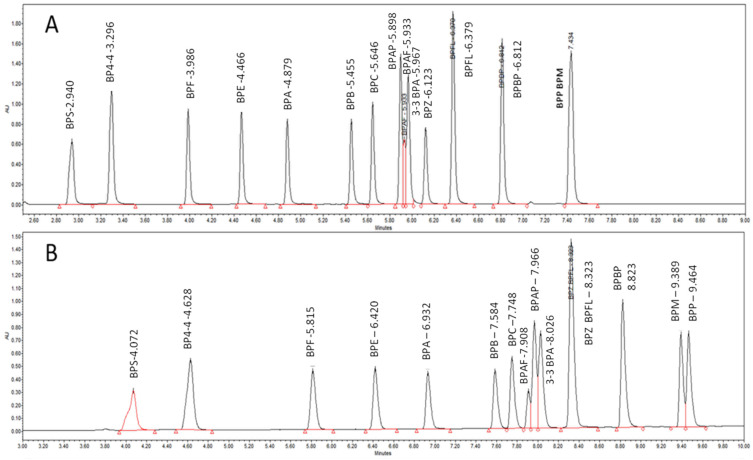
Chromatograms of 15 studied BPs eluted on a BEH C18 column with an AcN/H_2_O gradient elution (**A**) and with an MeOH/H_2_O gradient elution (**B**).

**Figure 3 molecules-28-00500-f003:**
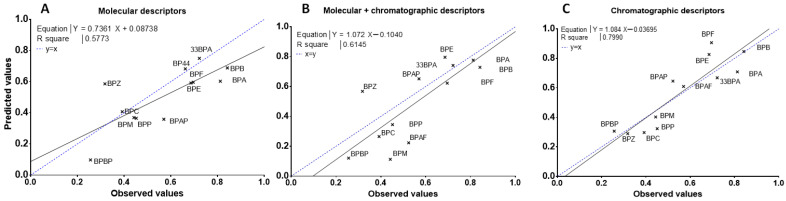
LMO validation curves of the mean observed placental CI of BPs and predicted for the three test data sets by multilinear regression based on molecular (**A**), both molecular and chromatographic (**B**), or chromatographic (**C**) descriptors. The dotted line represents a perfect fit with the equation CIobs = CIpred and the straight line is the representation of CIpred = a × CIobs + b where a and b are the coefficients of the corresponding linear regression model.

**Figure 4 molecules-28-00500-f004:**
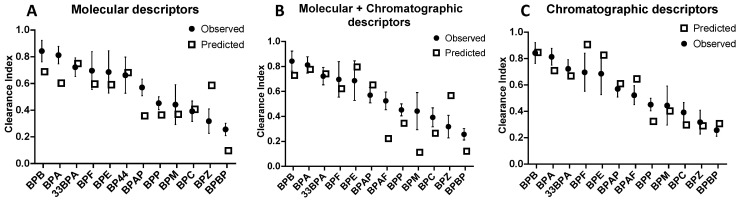
Mean ± SD observed CI determined for the BPs on five perfused human placentae compared to the CI predicted by QSAR models based on molecular descriptors (**A**), molecular combined with chromatographic descriptors (**B**), and chromatographic descriptors (**C**). The placental CI of some BPs cannot be predicted because they are pivotal in building the QSAR model (BPAF, BPFL, BPS in (**A**), BP44, BPFL, BPS in (**B**) and (**C**)).

**Table 1 molecules-28-00500-t001:** Mean ± standard deviation (SD) of CI of the 15 studied BPs (seven BPs with transfer rates that did not differ significantly from that of antipyrine and eight BPs with a significantly lower transfer rate than that of antipyrine) evaluated on five placentae. The clearance index corresponds to the ratio of BP-studied placental transfer rate divided by the antipyrine transfer rate [17].

Bisphenol	Mean CI ± SD (*n* = 5)	Classification
BPFL	0.064 ± 0.021	Significantly different from antipyrine transfer rate
BPS	0.082 ± 0.016	
BPBP	0.256 ± 0.046	
BPZ	0.318 ± 0.090	
BPC	0.392 ± 0.076	
BPM	0.442 ± 0.149	
BPP	0.452 ± 0.048	
BPAF	0.524 ± 0.071	
BPAP	0.570 ± 0.062	Not different from antipyrine transfer rate
BP4-4	0.662 ± 0.135	
BPE	0.686 ± 0.158	
BPF	0.696 ± 0.142	
3-3BPA	0.722 ± 0.070	
BPA	0.812 ± 0.065	
BPB	0.842 ± 0.080	

**Table 2 molecules-28-00500-t002:** Statistical results of the QSAR for the three selected models built with either molecular, chromatographic, or combined descriptors.

	Regression			Cross-Validation			Leave-Many-Out (LMO) Validation		
	RMSEC *	R^2^ *	BIC	RMSECV *	Q^2^ *	QLMO^2^ *	RMSEP **	CCC **	Rspearman **
Validation Criterion		> 0.65			> 0.5	> 0.65		> 0.85	
Molecular Descriptors	0.14	0.77	−64	0.14	0.71	0.57	0.17	0.64	0.7
Chromatographic Descriptors	0.11	0.84	−89	0.11	0.81	0.73	0.13	0.81	0.83
Both Descriptors	0.11	0.85	−86	0.11	0.82	0.47	0.19	0.67	0.71

* mean of validation parameters for the prediction of BPs of the three test data sets. ** parameters determined with observed and predicted CI calculated with test data set 1, 2, and 3.

**Table 3 molecules-28-00500-t003:** Molecular and chromatographic descriptors selected to build QSAR for predicting placental CI, i.e., with weight in the artificial neural networks higher than twice the median of all descriptors’ weight.

Molecular Descriptors		Chromatographic Descriptors			
Id	Parameters	Id	Column—Solvent—Parameters	Id	Column—Solvent—Parameters
CPS3	Connolly Solvent Excluded Volume	C18A2	C18—AcN—Width (5%)	RCBA2	RCB—AcN—Width (5%)
		C18A7	C18—AcN—Asymmetry	RCBA8	RCB—AcN—Tailing factor
CD1	Mol Refractivity	C18A8	C18—AcN—Tailing factor	CC18A2	CC18—AcN—Width (5%)
MT2	Cluster Count	PHA5	PH—AcN—k’ vs. BPA	CC18A7	CC18—AcN—Asymmetry
MT5	Polar Surface Area	FPA2	FP—AcN—Width (5%)	C18M5	C18—MeOH—k′ vs. BPA
MT12	Total Connectivity	C8A2	C8—AcN—Width (5%)	C18M7	C18—MeOH—Asymmetry
DE1	Heat of formation	T3A2	T3—AcN—Width (5%)	PHM5	PH—MeOH—k′ vs. BPA
DE2	Total Energy	T3A5	T3—AcN—k’ vs. BPA	FPM7	FP—MeOH—Asymmetry
DE5	Cosmo Area	T3A7	T3—AcN—Asymmetry	T3M5	T3—MeOH—k′ vs. BPA
DE9	Lumo Energy	T3A8	T3—AcN—Width (5%)	PFPM2	PFP—MeOH—Width (5%)
		RBA5	RB—AcN—k’ vs. BPA	PFPM7	PFP—MeOH—Asymmetry
		PFPA2	PFP—AcN—Width (5%)	RPM7	RP—MeOH—Asymmetry
		RPA2	RP—AcN—Width (5%)	CNM7	CN—MeOH—Asymmetry
		CNA5	CN—AcN—k’ vs. BPA	CC18M2	CC18—MeOH—Width (5%)
		FBA7	FB—AcN—Asymmetry		

**Table 4 molecules-28-00500-t004:** Chromatographic descriptors with weight in the artificial neural networks higher than twice the median of all descriptors’ weight and selected to build the QSAR model for predicting placental passage.

Chromatographic Descriptors			
Id	Column—Solvent—Parameters	Id	Column—Solvent—Parameters
T3A2	T3—AcN—Width (5%)	FPM7	FP—MeOH—Asymmetry
RBA7	RB—AcN—Asymmetry	C8M7	C8—MeOH—Asymmetry
PFPA7	PFP—AcN—Asymmetry	T3M5	T3—MeOH—k′ vs. BPA
FBA5	FB—AcN—k′ vs. BPA	PFPM7	PFP—MeOH—Asymmetry
CC18A2	CC18—AcN—Width (5%)	RPM8	RP—MeOH—Tailing factor
CC18A7	CC18—AcN—Asymmetry	CNM8	CN—MeOH—Tailing factor
C18M2	C18—MeOH—Width (5%)	FBM2	FB—MeOH—Width (5%)
C18M5	C18—MeOH—k′ vs. BPA	FBM7	FB—MeOH—Asymmetry
C18M7	C18—MeOH—Asymmetry		

**Table 5 molecules-28-00500-t005:** Analytical columns used for determining the chromatographic descriptors with their abbreviation, dimension, and selectivity [48,59].

Column	Dimension—Granulometry—Supplier	Selectivity
Raptor Biphenyl (RB)	100 × 2.1 mm; 2.7 µm, Restek	Polarizability, aromatic and dipolar selectivity
Raptor Biphenyl Core-Shell (RCB)	100 × 2.1 mm; 1.8 µm, Restek	Polarizability, aromatic and dipolar selectivity
Force Biphenyl (FB)	100 × 2.1 mm; 1.8 µm, Restek	Polarizability, aromatic and dipolar selectivity
Cortecs C18 (CC18)	100 × 2.1 mm; 1.6 µm, Waters	Hydrophobicity selectivity
BEH C18 (C18)	100 × 2.1 mm; 1.7 µm, Waters	Hydrophobicity selectivity (reference)
BEH RP 18 Shield (RP18)	100 × 2.1 mm; 1.7 µm, Waters	Basic compound selectivity
BEH C8 (C8)	100 × 2.1 mm; 1.7 µm, Waters	Hydrophobicity selectivity
BEH Phenyl (P)	100 × 2.1 mm; 1.7 µm, Waters	Pi-Pi selectivity
CSH Phenyl-Hexyl (PH)	100 × 2.1 mm; 1.7 µm, Waters	Pi-Pi selectivity
CSH Fluoro-Phenyl (FP)	100 × 2.1 mm; 1.7 µm, Waters	Halogenated and polar compound selectivity
HSS T3 (T3)	100 × 2.1 mm; 1.8 µm, Waters	Polar and hydrophobic molecule selectivity
HSS PFP (PFP)	100 × 2.1 mm; 1.8 µm, Waters	Pi-Pi, H-bonding, dipolar and hydrophobicity selectivity
HSS CN (CN)	100 × 2.1 mm; 1.8 µm, Waters	Alternative to hydrophobicity selectivity

## Data Availability

Molecular and chromatographic descriptors for the 15 bisphenols are described in the Excel file Appendix A.

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
