# Peer review of "Contribution of Reliable Chromatographic Data in QSAR for Modelling Bisphenol Transport across the Human Placenta Barrier"

_molecules, 2023, doi:10.3390/molecules28020500_

Round 1

Reviewer 1 Report

REVISION

In the manuscript entitled Contribution of reliable chromatographic data in QSAR for modelling bisphenol transport across the human placenta barrier, the authors implemented the set of chromatographic descriptors in three QSAR models and developed their ability to predict placental transfer of BPs compared to molecular descriptors. It is interesting that in this study, in addition to classical retention factors, chromatographic peak shape parameters were included as chromatographic descriptors in QSAR modeling. It was found that the model based only on chromatographic descriptors showed a better predictive ability. The existence of similarities between the interactions that the substance achieves with the mobile and stationary phases, with those that it achieves when passing through different physiological barriers, indicates the validity of the obtained results.

The manuscript is well written and organized.

For some future work, I would suggest to the authors to include other predictors (pharmacokinetic or toxicological) in QSAR modeling, as well as some non-commercial compounds of the investigated class, because the study will gain additional quality.

In my opinion, this paper has elements of scientific significance so I suggest acceptance of the manuscript in this form.

Author Response

We thank Reviewer 1 for his comments and suggestions that we will take into account in our future research.

Reviewer 2 Report

The manuscript presents transport data across human placenta barrier of  15 bisphenol deriatives modeled by QSAR equations which employed cohorts of structural and and chromatographic descriptors. The work is well designed and executed, the reviewer, however, have one comment to the general QSAR conclusions derived from the study: the authors constructed  equations with large number of independent variables (four in Eq. 1 and seven in Eq.2. and six variables in Eq. 3) to explain CI values derived for 15 compounds. These are surprinsingly high numbers given for number of compounds in the dependent variable, even when single CI measurement on each of five placenta models where included in the modeling. The authors should elaborate on that issue and provide additional statistical validation for proposed approach.       

Author Response

We thank Reviewer 2 for his comments, which are very relevant. We determined that the best models determined from five replicates per observation, include four independent variables in equation 1, seven in equation 2 and six in equation 3 to explain CI values of 15 observations. The number of parameters seems high for 15 observations with an apparent degree of freedom ranging from 6 to 9, however as described in the material and methods section §4.2.1 line 437, we were careful to check that the descriptors in each model were significant and uncorrelated. In addition, we performed the multilinear regression taking into account the experimental variability of the clearance indices leading to a degree of freedom of 67-70, which is much higher than the number of estimated parameters. We checked the number of parameters in the multilinear regression model by calculating the Bayesian information criterion for each model. It is clear that, despite the penalties increasing with the number of parameters, the model of Eq3 with six variables provides the lowest BIC value. We added this statistical validation in Table 2 and discussed it throughout the manuscript (lines 203-205, 235, 302-303 and 452-455).

Reviewer 3 Report

The article contains a lot of data and collected valuable results. The described experiments were very time-consuming and labor-intensive. This work contains too many abbreviations that are easy to confuse. The molecular structures (in Figure1) are not of the expected guality.

Author Response

We agree with Reviewer 3 regarding abbreviations.  Indeed, we generated a lot of data both with the descriptors and with the validation procedures. It was therefore difficult to limit the use of abbreviations; we tried as much as possible to report the most useful ones in tables and to describe them in the text. We made modifications throughout the manuscript (lines 131, 141, 147-151, 170, 199). 

As suggested, we modified figure 1 in the expected quality.